# The Effect of Modifying Canadian Goldenrod (*Solidago canadensis*) Biomass with Ammonia and Epichlorohydrin on the Sorption Efficiency of Anionic Dyes from Water Solutions

**DOI:** 10.3390/ma16134586

**Published:** 2023-06-25

**Authors:** Karolina Paczyńska, Tomasz Jóźwiak, Urszula Filipkowska

**Affiliations:** Department of Environmental Engineering, University of Warmia and Mazury in Olsztyn, Warszawska St. 117a, 10-957 Olsztyn, Poland; karolina.paczynska@student.uwm.edu.pl (K.P.); urszula.filipkowska@uwm.edu.pl (U.F.)

**Keywords:** Canadian goldenrod, biomass, amination, unconventional sorbent, sorption, dyes, Reactive Black 5, Reactive Yellow 84

## Abstract

This study examined the effect of modifying Canadian goldenrod (*Solidago canadensis*) biomass on its sorption capacity of Reactive Black 5 (RB5) and Reactive Yellow 84 anionic dyes. The scope of the research included the characteristics of sorbents (FTIR, elementary analysis, pH_PZC_), the effect of pH on dye sorption efficiency, sorption kinetics, and the maximum sorption capacity (describing the data with Langmuir 1 and 2 and Freundlich models). FTIR analyses showed the appearance of amine functional groups in the materials modified with ammonia water, which is indicative of the sorbent amination process. The amination efficiency was higher in the case of materials pre-activated with epichlorohydrin, which was confirmed by elemental analysis and pH_PZC_ values. The sorption efficiency of RB5 and RY84 on the tested sorbents was the highest in the pH range of 2–3. The sorption capacity of the goldenrod biomass pre-activated with epichlorohydrin and then aminated with ammonia water was 71.30 mg/g and 59.29 mg/g in the case of RB5 and RY84, respectively, and was higher by 2970% and 2510%, respectively, compared to the unmodified biomass. Amination of biomass pre-activated with epichlorohydrin can increase its sorption capacity, even by several dozen times.

## 1. Introduction

Post-production wastewater from the textile, tanning or paper industries may contain significant amounts of dyes. Industrial dyes are usually of synthetic origin, feature a complicated chemical structure, and are highly stable, which makes them sparingly biodegradable [1]. Due to the high color intensity and ease of application, anionic reactive dyes are most often used in dyeing processes [2]. For this reason, they are the most common type of contaminants found in colored industrial wastewater.

Colored substances discharged into natural waters can elicit a number of adverse effects in aquatic ecosystems. Even their low concentrations in water are very visible and disturb the aesthetics of water reservoirs [3]. A more serious problem, however, is that they substantially limit the access of the aquatic autotrophs to solar radiation, thereby disrupting their primary production in the local aquatic environment [4]. In addition, dyes can diminish atmospheric oxygen diffusion in water, which, when combined with the inhibited photosynthesis of hydrophytes, can lead to the development of anaerobic conditions in water reservoirs. A significant part of these dyes may also be toxic to the flora and fauna of lakes and rivers [5]. Bearing in mind the welfare of the natural environment, it is essential to minimize the risk of its contamination with dyes. Since contemporary civilization will not give up the production and use of dyes, it is necessary to use the most effective wastewater treatment technologies.

Due to the low biodegradability of most types of dyes, the treatment of colored wastewater with biological methods based on activated sludge or bio-membrane technology is usually ineffective [6]. It may, however, prove viable when performed with chemical or physical methods, including precipitation methods (coagulation, electrocoagulation), deep oxidation methods (ozonation, reaction with sodium hypochlorite), membrane methods (ultrafiltration, reverse osmosis), and sorption methods (adsorption, ionic exchange) [7]. The advantages of precipitation methods include a relatively short process time and over 90% efficiency, whereas their major drawback is the formation of significant amounts of hardly manageable precipitation sludge [8]. Decolorization of wastewater via dye oxidation is quick and easy to carry out, but does not prove effective enough for all groups of dyes. In addition, in systems based on chemical oxidation, there is a high risk of accumulation of intermediate oxidation products (e.g., aromatic amines), which are much more toxic than dyes [9]. Systems based on reverse osmosis ensure the highest efficiency of colored wastewater treatment [10]; however, they require expensive membranes and high pressures of up to 2.5 MPa to operate. Other disadvantages of reverse osmosis include high water losses, reaching up to 25%, as well as difficulties with the regeneration of worn membranes. By way of contrast, sorption methods offer the most environmentally friendly solution for wastewater decolorization, as they generate neither sludge nor toxic by-products. The sorption process is relatively simple to carry out and its costs depend mainly on the type of sorbent used [11]. 

The most common commercial sorbents include materials based on activated carbons, which are characterized by high porosity and a large specific surface area, reaching even over 1000 m^2^/g, due to which they effectively bind dyes from aqueous solutions [12]. The disadvantage of activated carbons is the high cost of their production and regeneration, which is why the search is underway for cheaper or more efficient substitutes.

Chitosan is the material that can compete with activated carbons in terms of dye sorption efficiency. It is a biopolymer, polysaccharide, and a deacetylated form of chitin, which is the main structural compound of joints of exoskeletons. Chitin and chitosan are produced on an industrial scale from crab and shrimp shells, both being forms of waste from the seafood processing industry [13]. Compared to the anionic reactive dyes popular in the industry, chitosan sorbents may have a sorption capacity several times higher than commercial activated carbons [14]. The high sorption capacity of chitosan is due to the basic amine functional groups characteristic of this polysaccharide, which serve as the key sorption centers for anionic dyes [15]. A drawback of chitosan as a sorbent is the low availability of the raw material in countries whose food industries do not include marine crustaceans. The availability of chitin and chitosan on the market is also limited, due to pharmaceutical companies that purchase these polysaccharides to produce dietary supplements. Much cheaper substitutes for activated carbons and chitosan are offered by waste materials based on plant biomass, e.g., from the agricultural and food industry. These include, for example, the stems and leaves of crops, vegetable and fruit peels, the shells of seeds and nuts, and cereal husks [16]. Besides their low price, the great advantage of these materials is their wide availability in almost every country worldwide. 

In addition to waste materials from agriculture, the biomass of invasive plants can also serve as a cheap and widely available source of biomass for sorbent production. An example of an invasive plant is Canadian goldenrod (*Solidago canadensis* L.) [17], which is a perennial plant belonging to the Asteraceae family and native to North America. In the 17th century, this species was brought to England as an ornamental plant, and had quickly spread throughout Eurasia by the end of the 20th century. It is characterized by a rapid biomass growth and high competitiveness with native species. After appearing in the ecosystem, Canadian goldenrod quickly displaces other plants, thereby significantly diminishing species diversity and leading to landscape homogenization [18], which is why it had been included in the so-called “blacklist of invasive species” of many European countries. Currently, the only way to control this plant is to mow it twice a year for several consecutive years. Theoretically, significant amounts of mown Canadian goldenrod biomass can be used as a material for sorbent production.

Plant biomass usually has relatively low sorption capacity with regard to anionic dyes. This is due to the generally acidic or neutral nature of plant biomass and, as a rule, the low content of basic functional groups in its structure [19]. Appropriate chemical modification of plant biomass can significantly increase its sorption capacity. This is the case with amination, which consists of incorporating primary amino groups [20,21] into the sorbent’s structure. As a result, the modified material, and likewise chitosan, gains an alkaline nature and a much higher capacity for anionic dye sorption [22]. An ammonia water bath offers the simplest but least effective method for biomass amination. However, the efficiency of material amination can be boosted by its preliminary chemical activation, e.g., with epichlorohydrin [23].

The present study investigated the possibility of enriching Canadian goldenrod biomass with amine functional groups and its effect on the sorption efficiency of popular industrial dyes, Reactive Black 5 and Reactive Yellow 84.

## 2. Materials and Methods

### 2.1. Canadian Goldenrod Biomass

The study was conducted with the biomass (stems with leaves and inflorescences) of Canadian goldenrod (*Solidago canadensis* L.) mown on 15 August 2022, in a meadow in the Warmia–Mazury Province (Poland). The average composition of its dry matter is as follows: cellulose (35–37%), hemicellulose (36–37%), lignin (18–19%), ashes/ash (2–3%), proteins and other components (4–9%) [24,25].

### 2.2. Dyes

The Reactive Black 5 (RB5) and Reactive Yellow 84 (RY84) dyes used in the study were provided by the dye-producing plant “Boruta” SA (Zgierz, Poland). Table 1 presents the key parameters of dyes provided by the producer.

### 2.3. Chemical Reagents

The following chemical reagents were used for analyses:Hydrochloric acid (HCl)—37%—(diluted, to correct the pH of dye solutions);Sodium hydroxide (NaOH) > 99.9% (microgranules)—(in solution form, for the correction of pH of solutions);Ammonia (NH_3_·H_2_O)—30%, 0.892 g/mL—(for the amination of goldenrod-based sorbent);Epichlorohydrin (C_3_H_5_ClO) > 99.0%—(for the modification/activation of straw-based sorbent).

All chemical reagents used were purchased from POCH S.A., (Gliwice, Poland) and were of analytical purity or higher.

### 2.4. Laboratory Equipment

The following laboratory equipment was used in the study:Laboratory grinder Microfine MF-2 (CONBEST, Kraków, Poland)—(for crushing the goldenrod biomass);Water bath shaker type 357 (Elpin-Plus, Lubawa, Poland)—(for the modification/activation of goldenrod biomass with epichlorohydrin);HI 110 pH meter (HANNA Instruments, Olsztyn, Poland)—(for the measurement and correction of the solutions’ pH);Laboratory shaker SK-71 (JEIO TECH, Daejeon, Korea)—(for the process of sorption);Multi-Channel Stirrer MS-53M (JEIO TECH, Daejeon, Korea)—for dye sorption analyses;UV-3100 PC—UV/Visible spectrophotometer (VWR spectrophotometers, VWR International LLC., Mississauga, ON, Canada)—(for determining the concentration of dye in solutions);FT/IR-4700LE FT-IR Spectrometer with a single reflection ATR attachment (JASCO International, Tokyo, Japan)—(for preparing the sorbent’s FTIR spectra);FLASH 2000 analyzer (Thermo Scientific, Waltham, MA, USA)—(for elemental analysis, and for the measurement of carbon and nitrogen contents).

### 2.5. Preparation of Sorbent Based on Goldenrod Biomass (GB)

First, the Canadian goldenrod biomass (stems with leaves and inflorescences) was rinsed with deionized water and then dried in a dryer at a temperature of 105 °C. The dried biomass was disintegrated in a laboratory grinder and then sieved through screens with mesh diameters of 3 mm and 2 mm. The biomass fraction 2–3 mm in diameter was again rinsed with deionized water. After re-drying, the goldenrod biomass (GB) was ready to be used in experiments.

### 2.6. Preparation of Aminated Goldenrod Biomass (GB-A)

The goldenrod biomass (GB, 50.0 g DM (DM—dry matter)) prepared following the procedures described in Section 2.5 was placed in a conical flask (vol. 1000 mL). Then, 400 mL of ammonia water (30%) was poured into the flask, which was then protected with a parafilm. The flask was placed on a shaker with a water bath under a fume cupboard (120 r.p.m., vibration amplitude 25 mm, temperature 25 °C) for 24 h. Afterwards, the biomass was filtered and washed with deionized water on a laboratory screen until the ammonia odor was no longer perceptible. After drying at 105 °C, the aminated goldenrod biomass (GB-A) was ready to be used in experiments.

### 2.7. Preparation of Goldenrod Biomass Modified with Epichlorohydrin (GB-E)

Goldenrod biomass (GB, 50.0 g DM) was placed in a conical flask (vol. 1000 mL) and poured with 400 mL of a 95% epichlorohydrin solution with pH = 12 (pH adjusted with NaOH). Afterwards, the flask was protected with a parafilm and placed on a shaker with a water bath (120 r.p.m., vibration amplitude 25 mm, temperature 60 °C). After 24 h modification, the biomass was filtered off and rinsed with a large volume of deionized water until the filtrate reached a neutral pH (pH < 7.5). After drying (105 °C), the goldenrod biomass activated with epichlorohydrin (GB-E) was ready to be used in experiments.

### 2.8. Preparation of Aminated Goldenrod Biomass Pre-Activated with Epichlorohydrin (GB-EA)

The biomass of goldenrod modified with epichlorohydrin (GB-E), prepared as described in Section 2.7, was aminated following the procedure described in Section 2.6.

All sorbents produced were stored in air-tight polyethylene containers. 

Figure 1 presents a simplified scheme of preparation of sorbents used in the study.

### 2.9. Analyses of pH Effect on Dye Sorption Efficiency

Briefly, 1.00 g DM of each sorbent was weighed, using a precise scale, into a series of conical flasks (300 mL). Then, 200-mL portions of previously prepared dye solutions with a concentration of 50 mg/L and pH 2–11 were poured into the flasks, which were placed on a multi-station laboratory shaker (150 r.p.m., 30 mm vibration amplitude) and shaken for 120 min. Afterwards, 10-mL portions of the samples were collected with an automatic pipette from the flasks to the signed polypropylene test tubes. The pH values of the dye solutions after sorption were also measured.

#### Measurement of pH_PZC_ Using the “Drift” Method

In researching the pH_PZC_ of sorbents, deionized water with pH correction in the pH range of 2–11 was used instead of dye solutions. After 2 h of mixing the sorbent in water solutions (pH 2–11), its pH was measured. A line chart was made (the X axis is the initial pH, and the Y axis is the difference between the final pH and the initial pH (pH_E_–pH_0_)) for each sorbent. The intersection of the line with the X axis denotes the pH_PZC_ point of the sorbent.

### 2.10. Analyses of Dye Sorption Kinetics

The sorbents were weighed in doses of 20.00 g DM to a series of beakers (2500 mL). Next, 2000 mL of dye solutions with concentrations of 50–200 mg/L and an optimal sorption pH (established as in Section 2.9.) were poured into the beakers. Afterwards, the beakers were placed on magnetic stirrers (200 r.p.m.), and their contents were mixed using standard magnetic stirrers (50 × 8 mm). Samples of the solutions (5 mL each) were collected using an automatic pipette at intervals of 0, 10, 20, 30, 45, 60, 90, 120, 150, 180, 210, 240, 270, and 300 min into the previously prepared test tubes.

### 2.11. Analyses of the Maximal Sorption Capacity of the Sorbents Used in the Study

Portions of sorbents (1.00 g DM) were weighed into a series of conical flasks (300 mL). Then, 200 mL of dye solutions with concentrations of 10–250 mg/L (for GB, GB-A, GB-E) or 10–500 mg/L (for GB-EA) were poured into the flasks. The dye solutions had an optimal sorption pH (established following the procedures described in Section 2.9). The flasks were placed on a shaker (150 r.p.m., vibration amplitude 30 mm) for the time needed to reach sorption equilibrium (determined using the methodology described in Section 2.10). Afterwards, 10-mL samples of dye solutions were collected from the beakers into the earlier prepared test tubes.

The sample preparation procedures described in Section 2.9, Section 2.10 and Section 2.11 were performed at the minimal stirring rate, ensuring equal sorbent distribution throughout the entire volume of solution. The concentrations of dyes in the samples were determined with the spectrophotometric method using a UV-VIS spectrophotometer.

All experimental series were performed in triplicate.

### 2.12. Computation Methods

The amount of dye bound to the sorbent was computed using Equation (1):(1)$$\mathrm{Qs}=\left(C_{0}-C_{S}\right)\times \frac{V}{m}$$

Q_S_—mass of sorbed dye [mg/g];

C_0_—initial concentration of the dye [mg/L];

C_s_—concentration of the dye after sorption [mg/L]; 

V—volume of the solution [L]; 

m—sorbent mass [g].

Experimental data obtained from studies into the kinetics of dye sorption onto the tested sorbents were described using the pseudo-first-order model (2), pseudo-second-order model (3), and intraparticle diffusion model (4).
(2)$$Q=q_{e}\times \left(1-e^{\left(-k_{1}\times t\right)}\right)$$
(3)$$Q=\frac{\left(k_{2}\times {\;q}_{e}{}^{2}\times \;t\right)}{\left(1+{\;k}_{2}\times {\;q}_{e}\times \;t\right)}$$
(4)$$Q=k_{\mathrm{id}}\times t^{0.5}$$

Q—instantaneous value of sorbed dye [mg/g];

q_e_—the amount of dye sorbed at equilibrium state [mg/g];

t—time of sorption [min];

k_1_—pseudo-first-order adsorption rate constant [1/min];

k_2_—pseudo-second-order adsorption rate constant [g/(mg·min)];

k_id_—intraparticle diffusion model adsorption rate constant [mg/(g·min^0.5^)].

Experimental data obtained from studies into the maximal sorption capacity of the tested sorbents were described using Langmuir 1 (5), Langmuir 2 (double-Langmuir isotherm) (6), and Freundlich (7) isotherms.
(5)$$Q=\frac{\left(Q_{\max}\times {\;K}_{C}\times \;C\right)}{\left(1+{\;K}_{C}\times \;C\right)}$$
(6)$$Q=\frac{\left(b_{1}\times {\;K}_{1}\times \;C\right)}{\left(1+{\;K}_{1}\times \;C\right)}+\frac{\left(b_{2}\times {\;K}_{2}\times \;C\right)}{\left(1+{\;K}_{2}\times \;C\right)}$$
(7)$$Q=K\times C^{\frac{1}{n}}$$

Q—mass of sorbed dye [mg/g];

C—concentration of the dye left in the solution [mg/L];

Q_max_—maximum sorption capacity in Langmuir equation [mg/g];

b_1_—maximum sorption capacity of the sorbent (type I active sites) [mg/g];

b_2_—maximum sorption capacity of the sorbent (type II active sites) [mg/g];

K_C_—constant in Langmuir equation [L/mg];

K_1_; K_2_—constants in Langmuir 2 equation [L/mg]; 

K—the sorption equilibrium constant in the Freundlich model;

n—constant in the Freundlich model.

## 3. Results and Discussion

### 3.1. The FTIR Spectra Analysis and C/N Analysis of the Tested Sorbents

The FTIR spectrum of Canadian goldenrod biomass was typical of a lignocellulosic plant material (Figure 2). A wide band noticeable in each spectrum at 3600–3000 cm^−1^ denoted the stretching of the O-H bond of the hydroxyl functional groups. The peak visible at 3332 cm^−1^ in this band indicated the stretching of the N-H bond (amide A) of biomass proteins [26].

Small peaks visible at 2849 cm^−1^ and 2919 cm^−1^ may be ascribed to the hydrophobic symmetric and asymmetric stretching vibrations of CH_2_ [27]. In turn, peaks at 1509 cm^−1^, 1599 cm^−1^, and 1630 cm^−1^ correspond to the stretching of the C=C bond of the aromatic ring of GB lignin [28]. Also typical of the lignin benzene rings are peaks at 1424 cm^−1^ and 1458 cm^−1^, which correspond to the stretching of C-H bonds [29]. The peaks visible at 1156 cm^−1^, 1109 cm^−1^, 1025 cm^−1^, and also 899 cm^−1^ indicate the presence of the C-O-C bond of the saccharide rings of cellulose and hemicellulose, which are components of goldenrod biomass [30,31]. The peak at 1360 cm^−1^ (C-H bond vibration) and the one at 1320 cm^−1^, corresponding to the stretching of the C-O bond at the C5 carbon atom of the aromatic ring of cellulose or hemicellulose, are also typical of saccharides (Figure 2).

The peak visible at 1730 cm^−1^ only in the GB and GB-E spectra indicates the presence of the C=O bond, which is likely to belong to various functional groups of biomass lignins and hemicelluloses (carbonyl, ester, ketone, and carboxyl groups) [28]. In addition to the carbonyl group peak, GB and GB-E also have a distinct peak at 1250 cm^−1^, corresponding to the stretching vibrations of the -OH bond of the carboxyl group. The lack of these peaks in the GB-A and GB-EA spectra may suggest that the carboxyl and carbonyl groups entered into reactions with ammonia during the chemical modification of the test material. Instead of a distinct peak corresponding to the hydroxyl group (1250 cm^−1^), the GB-A and GB-EA spectra show two small peaks at 1268 cm^−1^ and 1240 cm^−1^, corresponding to the bending of the N-H bond and the stretching of the C-N bond [32].

In turn, the BG-E spectrum possesses characteristic peaks at 833 cm^−1^ and 860 cm^−1^, pointing to the presence of epoxide rings in the material [31], which confirms the reaction of goldenrod biomass with epichlorohydrin. The peak visible at 860 cm^−1^ was also present in the GB-EA spectrum, which may suggest that not all epoxide groups were involved in biomass amination (Figure 2).

An elementary analysis of the tested sorbents demonstrated that both the percentage content of nitrogen and the N/C ratio increased, respectively in the following sorbents: GB-EA > GB-A > GB > GB-E (Table 2). In turn, the nitrogen content of GB-A and GB-EA was 1.8% and 6.0% higher compared to that of GB. The above findings corroborate the higher efficiency of biomass amination in the case of its pre-activation with epichlorohydrin. The lower N/C ratio determined in GB-E was due to its higher carbon content resulting from the attachment of the epoxide group to its structure.

### 3.2. The Effect of pH on Dye Sorption Efficiency

The efficiency of RB5 and RY84’s sorption onto GB, GB-A, and GB-E was the highest at pH 2, whereas on GB-EA, the efficiency was highest at pH 3 (Figure 3a,b). In general, the pH increase in the system diminished the intensity of the dye’s binding onto sorbents. In the case of GB, GB-A, and GB-E, a substantial decrease was noted in RB5 and RY84 sorption efficiency from pH 2–5. Unlike the other sorbents, GB-EA ensured a high sorption efficiency within a broad pH range, i.e., from pH 2–10. In the experimental series with RB5 sorption onto the modified sorbents (GB-A, GB-E, GB-EA), a small increase was noted in the sorption intensity at pH 9. The intensity of RB5 and RY84 sorption onto each sorbent tested was the lowest at pH 11. 

The high efficiency of RB4 and RY84 sorption onto the analyzed sorbents at low pH values was due to the positive charge gained by sorbent’s surface. A significant excess of hydronium ions in the system led to the intensive protonation of hydroxyl or amine groups on the surface of sorbents.
-OH + H_3_O^+^ → -OH_2_^+^ + H_2_O 
-NH_2_ + H_3_O^+^ → -NH_3_^+^ + H_2_O 

The positively charged functional groups interacted electrostatically with anionic dyes, which substantially aided their sorption (Figure 3a,b). 

Usually, hydroxyl groups are the main functional groups responsible for physical adsorption on highly cellulosic plant biomass. Because the hydroxyl groups of polysaccharides and lignins undergo protonation only at a very low pH (pH 2–3) [33], the pH increase in the system caused a significant decrease in the number of ionized -OH groups, thus reducing the positive charge on the sorbent’s surface. These changes explain the significant decrease in RB5 and RY84 sorption efficiency noted at pH 2–5 in the case of GB, GB-A, and GB-E. Apart from hydroxyl groups, GB-EA also possesses multiple amine functional groups on its surface, as corroborated in Section 3.1. Most of the primary amine functional groups occur in the protonated form already at pH < 9. This would explain the high RB5 and RY84 sorption efficiency onto GB-EA within a broad pH range (Figure 3a,b). 

When there is a high pH in the system (pH >10), the functional groups (e.g., hydroxyl ones) are probably deprotonated on the surface of sorbents, thus gaining a negative charge (-OH + OH^−^ → -O^−^ + H_2_O). This caused the electrostatic repulsion of anionic dyes, which in turn resulted in a significant inhibition of the sorption process.

The positive impact of low pH on the sorption efficiency of anionic dyes was also observed in studies addressing RB5 sorption onto activated carbon [34], carbon nanotubes [35], buckwheat husks [23], and chitin-based sorbents [36]. The same tendency was also noted in research investigating RY84 sorption onto coconut shells [37], sunflower husks [22], compost [38], and chitosan-based sorbents [39,40].

A negligible increase noted in the RB5 sorption efficiency at pH 9 (Figure 3a) was probably due to the presence of the -NH_2_ groups in the dye’s structure. At pH 9, the sorbent’s surface presumably had already possessed a total negative charge, whereas a significant proportion of the RB5 particles still possessed a local positive charge in the form of an ionized -NH_3_^+^ group. The electrostatic interaction between the negatively charged surface of the sorbents and the protonated amine group of the dye could aid its sorption. At pH > 9, most amine groups of RB5 were already in the non-ionized form (-NH_2_), which arrested the electrostatic interaction of the sorbent with the dye. The increase noted in the RB5 sorption efficiency at pH 9 was also observed in research addressing its sorption onto sunflower husks [22], cotton fibers [21], and egg shell membranes [41].

The sorbents tested in the present study contributed to a significant change in the dye solution’s pH during sorption (Figure 3c,d). In the experimental series with an initial pH range of pH 5–10, the range of pH values noted after 120 min sorption was pH 6.83–7.30 for GB, pH 7.08–7.80 for GB-A, pH 6.39–7.15 for GB-E, and pH 8.03–8.39 for GB-EA. Changes in pH values observed in solutions containing sorbents are always due to the system’s tendency to reach a pH value close to the pH_PZC_ of the sorption material. The pH_PZC_ values determined with the “drift” method reached pH_PZC_ = 7.22 for GB, pH_PZC_ = 7.46 for GB-A, pH_PZC_ = 6.79 for GB-E, and pH_PZC_ = 8.25 for GB-EA (Figure 3e,f). GB-A and GB-EA have higher pH_PZC_ values due to the higher number of amine functional groups of an alkaline nature on the sorbent’s surface. This finding provides more evidence for the amination of the chemical structure of sorbents, which occurred during their bath in ammonia water. The significantly higher pH_PZC_ value of GB-EA compared to that of GB-A confirms the higher efficiency of the amination of the biomass pre-activated with epichlorohydrin. In turn, the lower pH_PZC_ value of GB-E compared to that of GB may stem from the formation of chloride ions upon epichlorohydrin’s reaction with the biomass. Thus, the formed Cl^−^ ions that were bound on GB-E’s surface could penetrate into the dye solution during sorption as a result of, e.g., ionic exchange. Consequently, the appearance of chloride ions in the system caused the solution’s pH, ultimately, to decrease (Figure 3e,f).

The efficiency of RB5 and RY84’s sorption onto the tested sorbents was the highest in the pH range of pH 2–3. However, owing to the fact that the pH values of colored industrial wastewater containing anionic dyes are usually higher than pH 2 [42], further experiments, described in Section 3.3 and Section 3.4, were performed at pH 3.

### 3.3. Dye Sorption Kinetics

The equilibrium time of RB5 and RY84’s sorption onto the analyzed sorbents depended on the sorbent type, dye type, and initial dye concentration (Table 3, Figure 4). In the experimental series with GB, GB-A, and GB-E, the sorption of RB5 spanned from 120 to 150 min, whereas that of RY84 took from 150 to 180 min. The dyes’ sorption onto GB-EA was more brief, and fell within the range of 90–120 min in the case of RB5 and 60–120 min in the case of RY84. For each sorbent tested, shorter equilibrium times were reached with higher initial concentrations of dyes. 

Similar equilibrium times of RB5 sorption were noted in studies addressing its sorption onto the seed hulls of Eriobotrya japonica (150 min) [43] and activated carbon from the carob tree (120 min) [44]. In the case of RY84, similar equilibrium sorption times were reported during dye removal onto wool (180 min) [45] and compost (180 min) [38].

Generally, the longer equilibrium times of RY84’s sorption compared to RB5’s sorption may be due to its higher molecular mass (RY84—1628 g/mol; RB5—991 g/mol). Larger sorbate molecules could impede and consequently delay the dyes’ binding to the active sites located in deeper layers of the sorbent [36]. The shorter sorption times noted in the experimental series with the higher initial concentrations of dyes could have been due to greater likelihood of collisions between sorbate molecules and the sorption centers of the sorbent. The faster saturation of active sites in the structure of the sorptive material was reflected in the earlier completion of the sorption process. A shorter sorption time at higher initial concentrations of the dye was also noted in studies on RB5’s sorption onto feathers [46] and pumpkin seed husks [47]. The shorter equilibrium times of dye sorption onto GB-EA compared with GB, GB-A, and GB-E stemmed from a high concentration of -NH_2_ groups on the sorbent’s surface. The amine functional groups of GB-EA, protonated at pH 3, were responsible for a strong positive charge on the sorbent’s surface, which intensified and accelerated the dyes’ binding from the solution [20,21,23]. No similar effect was observed in the case of GB-A (Figure 4), due to a substantially lower content of primary amine functional groups.

In all experimental series, the analytical data were best described with the pseudo-second-order model, regardless of the sorbent and dye type, which was found to be typical of the sorption of dyes onto biosorbents [48,49]. The k_2_ constants and q_e_ values determined using this model confirm the higher rate and intensity of dye sorption noticeable in Figure 4 for the experimental series with higher initial concentrations of dyes. 

Experimental data from analyses of the dye sorption kinetics were also described using the intraparticle diffusion model (Table 4, Figure 5). The results of the analyses presented in Figure 5 indicate that the sorption of RB5 and RY84 dyes onto all tested sorbents proceeded in two main phases.

The first phase of sorption was highly intensive but short. Presumably, in this phase, dye molecules diffused from the solution onto the sorbent surface and attached to the most accessible sorption centers [36]. The second phase probably began when most of the active sites of the sorbent’s surface had been saturated. This phase was less intensive but substantially longer than the first one, and consisted of the saturation of the last free sorption centers on sorbent’s surface [20]. In addition, strong competition was observed in this phase between dye molecules for the last available sorption centers. Interactions between dyes strongly diminished access to the active sites in the deeper layers of the material, thereby contributing to a significant elongation of the time needed by the system to reach sorption equilibrium.

The first, key phase of sorption was the shortest in the experimental series with GB-EA (Table 4). The values of the k_d1_ constants determined from the intraparticle diffusion model also confirm that the first phase of dye sorption onto GB-EA was more intensive than on the other sorbents tested. As already mentioned, the high rate of RB5 and RY84’s sorption onto GB-EA is presumably due to the high number of amine functional groups possessed by the sorbent, which, when protonated, significantly aided dye binding [21]. Because of the small number of -NH_2_ groups, a similar effect was not observed for GB-A.

Generally, the shorter duration of the first sorption phase of RY84 compared to RB5 (Table 4) may stem from its higher molar mass. The larger molecules of RY84 occupied available surface of the sorbent within a shorter time span and rapidly began to compete with each for free sorption centers; this was in turn reflected in the faster onset of the second phase of sorption. As already mentioned, it was more difficult for the larger RY84 molecules to occupy the active sites in the sorbent’s deeper layers in the second phase, which caused a delay in reaching sorption equilibrium.

Goldenrod biomass modification with epichlorohydrin had no significant effect on the dye sorption equilibrium time or on the duration of both sorption phases. Most likely, the rate of anionic dye sorption is affected to the greatest extent by the charge on the sorbent’s surface. The epoxide groups incorporated into the GB-E structure are not easily ionized [50], which results in their weak electrostatic interactions with the dye, and a sorption time similar to that of GB.

### 3.4. Maximal Sorption Capacity

Experimental data from analyses of the maximal sorption capacity were described using three popular sorption isotherms: Langmuir 1, Langmuir 2 (dual-site Langmuir), and Freundlich (Figure 6, Table 5). In each experimental series, the Langmuir 1 and 2 models showed a better fit to the experimental data, compared to the Freundlich model. This finding suggests that only one dye molecule could attach to one sorption center during the sorption process, which resulted in the formation of a dye “monolayer” on the sorbent’s surface.

In the case of the experimental series with GB, GB-A, and GB-EA, the Langmuir 1 and 2 models showed the same extent of fit to the experimental data (as indicated by the same value of determination coefficient, R^2^), and both the Q_max_ and K_C_/K_1_/K_2_ constants determined from these models had the same numeric values (Table 5). This suggests that only one type of active site played a key role during dye sorption onto these sorbents. Presumably, protonated hydroxyl groups (-OH_2_^+^) served this function for GB, whereas protonated amine groups (-NH_3_^+^) did so for GB-A and GB-EA. A similar result (the same values of R^2^, Q_max_, K_C_/K_1_/K_2_) was also obtained in research into RB5’s sorption onto feathers [46] and onto aminated wheat straw [20].

The Langmuir 2 model showed a better fit to the experimental data than the Langmuir 1 model in the experimental series with GB-E (Table 5), which indicates that at least two active sites played the major role during dye sorption onto this sorbent. Presumably, the two types of active site on GB-E were protonated hydroxyl groups (-OH_2_^+^) and epoxide groups.

The maximal RB5 sorption capacity of GB, GB-A, GB-E, and GB-EA reached 2.32 mg/g, 10.62 mg/g, 8.17 mg/g, and 71.30 mg/g, respectively. In the case of RY84, their sorption capacity was negligibly lower, and reached 2.27 mg/g, 8.85 mg/g, 6.93 mg/g, and 59.28 mg/g, respectively. These lower RY84 sorption capacities of the analyzed sorbents could be due to the larger size of the dye molecules, which made it difficult for them to reach the sorption centers located in the sorbent’s deeper layers.

In most experimental series, the values of K_C_/K_1_/K_2_ constants determined from the Langmuir 1 and Langmuir 2 models, indicating the extent of the sorbate’s affinity to the active sites of the sorbent, fell within a similar range (0.02–0.03). This may point to a similar mechanism of sorption, mainly involving interactions between the ionized groups of the dye (e.g., -SO_3_^−^) and the protonated functional groups of the sorbent (-OH_2_^+^/-NH_3_^+^) [51].

The RB5 and RY84 sorption efficiency of the tested sorbents, estimated based on results from Table 3 and Table 4 (point 3.3) and Table 5, could be ordered as follows: GB-EA > GB-A > GB-E > GB.

The low efficiency of dye sorption onto GB is due to the fact that the sorbents mainly possess -OH groups on their surface. The efficiency of the hydroxyl groups’ protonation is low at pH 3 [21,52], resulting in a small positive charge on the sorbent’s surface and, consequently, a small amount of dye being bound. The higher efficiency of GB-E compared to GB is due to the presence of epoxide groups on the surface of biomass modified with epichlorohydrin. The reactive epoxide groups could have been reacted with the functional groups of the dyes [53]. As a result, chemisorption could have aided the process of the physical adsorption of dyes onto GB-E. However, the chemisorption efficiency of dyes onto GB-E was lower compared to the efficiency of their physical adsorption onto the surface of sorbents containing protonated amine functional groups (GB-A, GB-EA). The highest sorption efficiency of RB5 and RY84 onto GB-EA was due to the fact that it possessed the highest number of easily ionizable -NH_2_ groups, which, as already mentioned, serve as the key sorption center of dyes. Due to the low amination efficiency of biomass not pre-activated with epichlorohydrin, GB-A had a much lower content of amino groups than GB-EA, which resulted in its much lower sorption efficiency.

Table 6 compares RB5 and RY84’s sorption efficiency on various biosorbents and activated carbons.

Aminated goldenrod biomass pre-activated with epichlorohydrin has similar sorption properties to other plant sorbents (wheat straw, buckwheat hulls, sunflower hulls) subjected to the same modifications (activation with epichlorohydrin + amination). Thus, modified plant biosorbents gain a many times higher sorption capacity compared to the unmodified plant biomass (Table 6). Generally, sorbents prepared following this method exhibit a similar or even higher sorption capacity than some activated carbons. This proves the high usefulness of plant material enrichment with the amine functional groups described in this paper. A prerequisite for obtaining high sorption capacity as a result of amination is biosorbent pre-activation with epichlorohydrin. As in the case of goldenrod biomass, the amination of straw or buckwheat hulls without their pre-modification with epichlorohydrin was less efficient, and resulted in only a slight increase in their sorption capacity (Table 6).

## 4. Conclusions

Amination of Canadian goldenrod biomass can significantly increase its sorption capacity towards anionic dyes. GB amination with an ammonia water solution proves much more effective in the case of materials pre-modified with epichlorohydrin. Goldenrod biomass pre-activated with epichlorohydrin and then aminated showed a 2510–2970% higher sorption capacity than the unmodified biomass, and could compete in this respect with some types of activated carbons.

The pH of the solution had a great impact on the efficiency of RB5 and RY84’s sorption onto the tested sorbents. The dye binding efficiency on GB, GB-A, GB-E, and GB-EA was the highest in the pH range of 2–3.

Sorbents based on goldenrod biomass modified the pH value of the solution in which sorption took place, which was due to the system’s tendency to reach a pH value similar to the pH_PZC_ of the sorbent used. The pH_PZC_ values determined for GB, GB-A, GB-E, and GB-EA were 7.22, 7.46, 6.79, and 8.25, respectively. Increased pH_PZC_ values of GB-A and GB-EA compared to those determined for GB resulted from the addition of basic -NH_2_ groups to the sorbent’s structure during amination.

The equilibrium time of dye sorption estimated for GB, GB-A, and GB-E ranged from 120 to 180 min, and that determined for GB-EA from 60 to 120 min. The shorter equilibrium times of dye sorption onto GB-EA than onto GB, GB-A, and GB-E stemmed from a high concentration of -NH_2_ groups on sorbent’s surface. Amine functional groups protonated at low pH were responsible for a strong positive charge on the sorbent’s surface, which accelerated RB5 and RY84 binding from the solution.

The sorption of anionic dyes onto GB, GB-A, GB-E, and GB-EA proceeded in two main phases. The first phase was characterized by a short duration (10–45 min) and high sorption efficiency, while the second phase was generally longer (50–130 min) and relatively less intensive.

It is likely that one type of active site played the key role in RB5 and RY84’s sorption onto GB, GB-A and GB-EA. These may be protonated hydroxyl groups for GB and protonated amino groups for GB-A and GB-EA. In the case of GB-E, the binding of dyes takes place on at least two active sites, probably protonated hydroxyl groups and epoxide groups.

## Figures and Tables

**Figure 1 materials-16-04586-f001:**
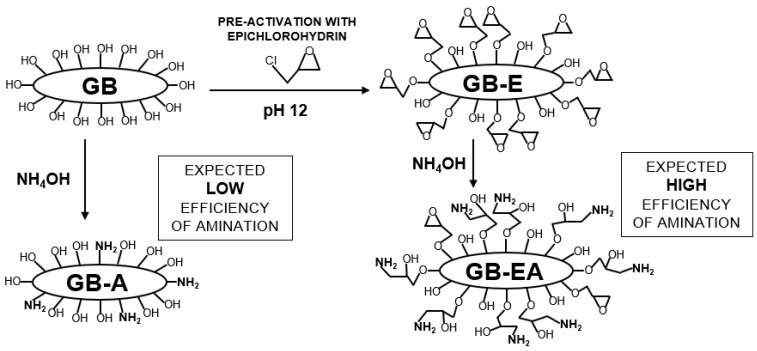
Scheme of preparation of GB, GB-A, GB-E, and GB-EA.

**Figure 2 materials-16-04586-f002:**
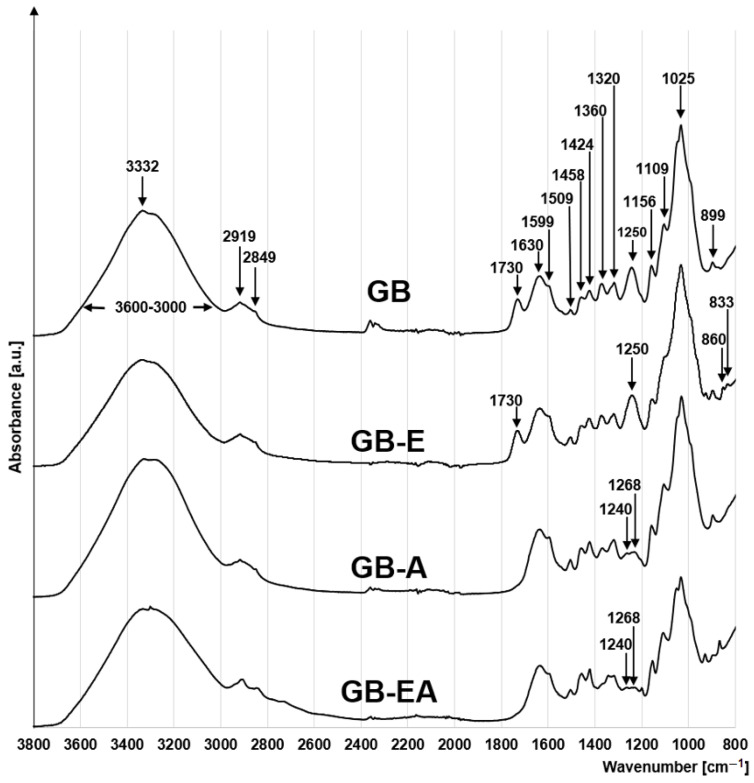
FTIR spectra for GB, GB-E, GB-A and GB-EA.

**Figure 3 materials-16-04586-f003:**
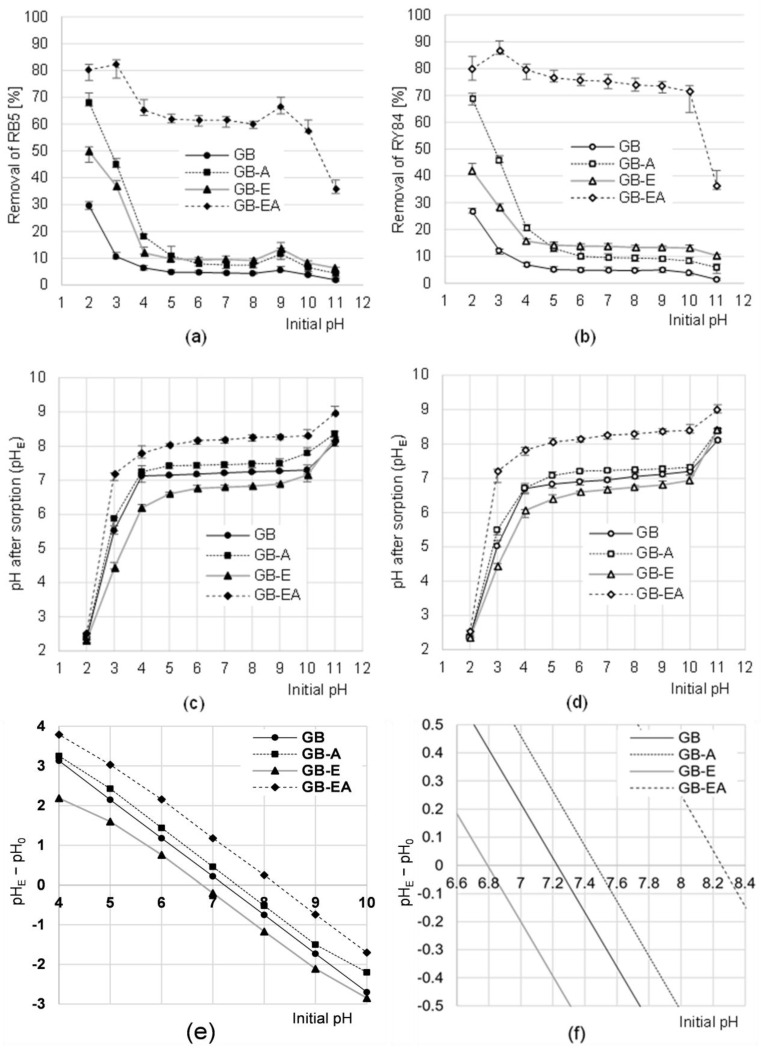
Effect of pH on the efficiency of sorption of (**a**) RB5 and (**b**) RY84 onto the tested sorbents (average + range). Effect of sorbents on changes in solution pH after sorption of (**c**) RB5 and (**d**) RY84. (**e**,**f**) Determination of pH_PZC_ of the tested sorbents using the “drift” method; temp. 22 °C.

**Figure 4 materials-16-04586-f004:**
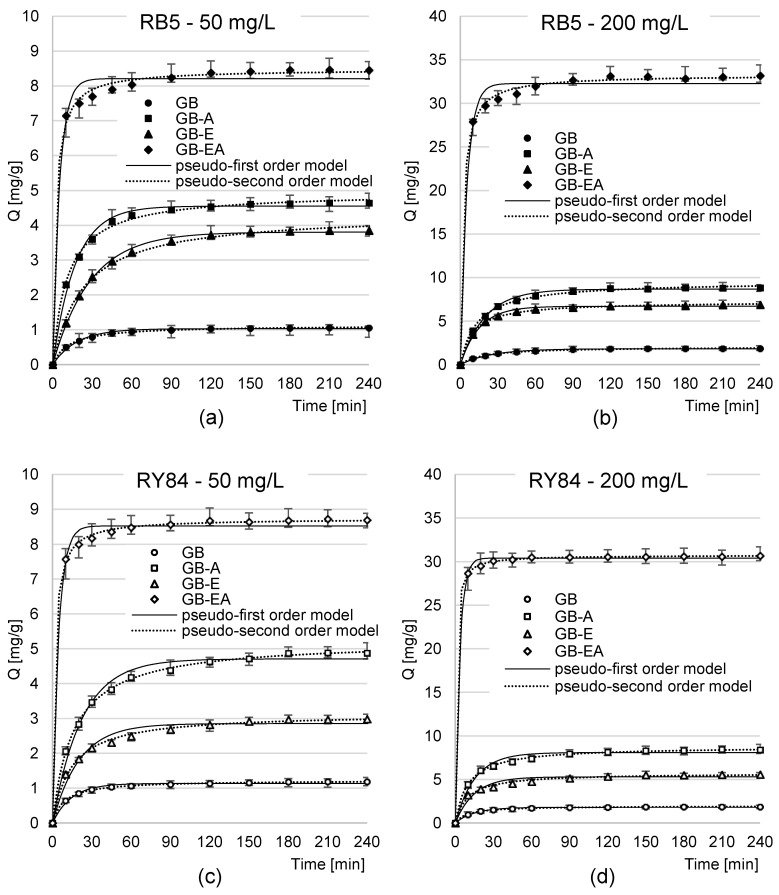
Changes in the concentration of dyes, (**a**) RB5—50 mg/L, (**b**) RB5—200 mg/L, (**c**) RY84—50 mg/L, and (**d**) RY84—200 mg/L, during sorption onto tested sorbents; temp. 22 °C.

**Figure 5 materials-16-04586-f005:**
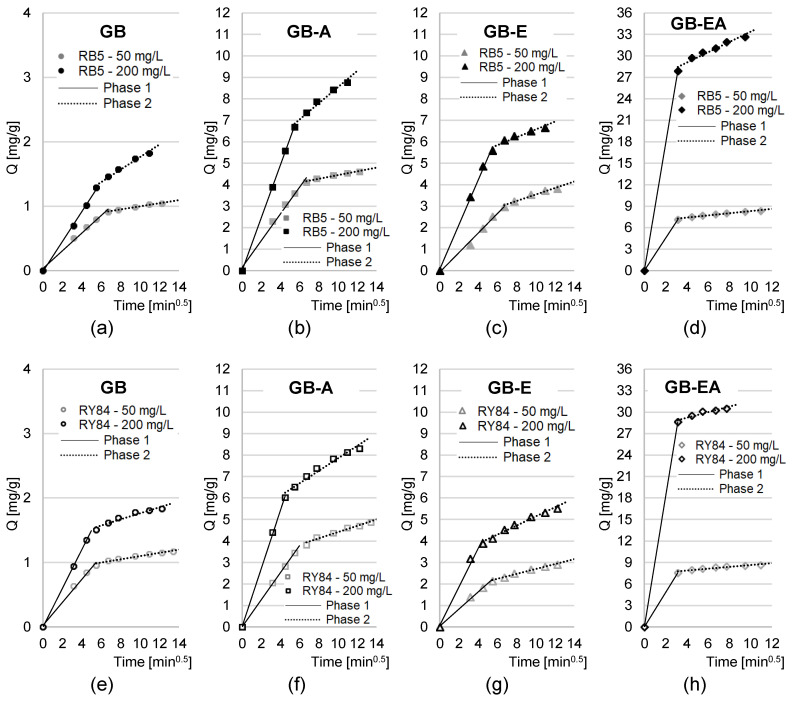
The intraparticle diffusion model of the sorption of RB5 onto (**a**) GB, (**b**) GB-A, (**c**) GB-E, (**d**) GB-EA, and RY84 onto (**e**) GB, (**f**) GB-A, (**g**) GB-E, and (**h**) GB-EA; temp. 22 °C.

**Figure 6 materials-16-04586-f006:**
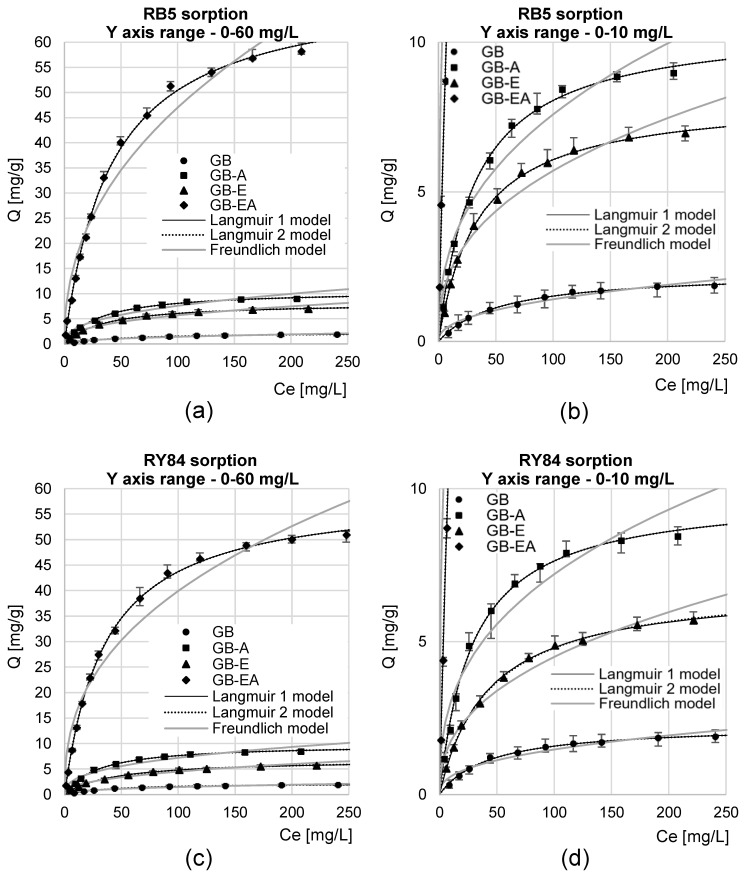
Isotherms of sorption of (**a**,**b**) RB5 and (**c**,**d**) RY84 onto the tested sorbents; temp. 25 °C.

**Table 1 materials-16-04586-t001:** Characteristics of Reactive Black 5 and Reactive Yellow 84 dyes.

Dye Name	Reactive Black 5 (RB5)	Reactive Yellow 84 (RY84)
Structural formula	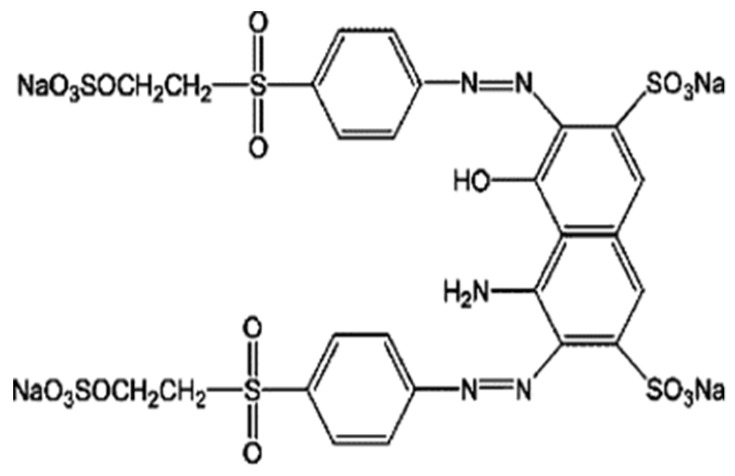	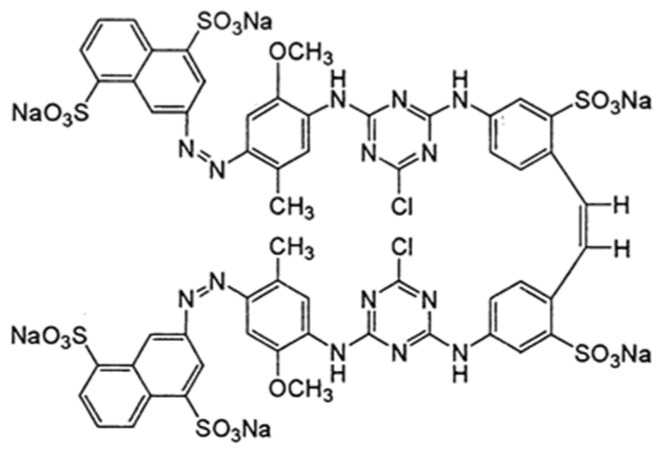
Chemical formula	C_26_H_21_N_5_Na_4_O_19_S_6_	C_56_H_38_Cl_2_N_14_Na_6_O_20_S_6_
Dye type	anionic (reactive)	anionic (reactive)
Class of dye	double azo	double azo
Type of active groups	vinylsulfone	chlorotriazine
Molecular weight	991 [g/mol]	1628 [g/mol]
Solubility in water	200 g/L (20 °C)	80 g/L (20 °C)
λ_max_	600 [nm]	356 [nm]
Dye application	dyeing of cotton, viscose, wool, polyamide fibers	dyeing of polyester, cotton, synthetic silk
Product purity	dye content 70%	dye content 97%

**Table 2 materials-16-04586-t002:** Carbon and nitrogen content in the tested sorbents. Analyses carried out on an elemental analyzer, FLASH 2000 (THERMO SCIENTIFIC, Waltham, MA, USA) (with three repetitions of measurements).

Type of Sorbent	Carbon Content [%]		Nitrogen Content [%]		N/C Ratio
	Range of Content	Average Content	Range of Content	Average Content	Based on Average Content
GB	46.24–46.41	46.533	1.006–1.016	1.012	0.0217
GB-A	46.77–46.95	46.362	1.025–1.037	1.030	0.0222
GB-E	47.42–47.71	47.564	1.005–1.014	1.009	0.0212
GB-EA	47.42–47.60	47.540	1.069–1.078	1.073	0.0226

**Table 3 materials-16-04586-t003:** Kinetic parameters of sorption of RB5 and RY84 onto GB, GB-A, GB-E, and GB-EA, determined from pseudo-first-order and pseudo-second-order models (based on the average of three measurements) + sorption equilibrium time.

Sorbent	Dye	Dye Conc.	Pseudo-First-Order Model			Pseudo-Second-Order Model			Exp. Data	Equil. Time
			k_1_	q_e,cal._	R^2^	k_2_	q_e,cal._	R^2^	q_e,exp._	
		[mg/L]	[1/min]	[mg/g]	-	[g/mg*min]	[mg/g]	-	[mg/g]	[min]
GB	RB5	50	0.0541	1.028	0.9888	0.0723	1.125	0.9984	1.052	150
		200	0.0398	1.822	0.9937	0.0263	2.046	0.9966	1.850	120
	RY84	50	0.0693	1.137	0.9848	0.0901	1.227	0.9994	1.185	180
		200	0.0663	1.802	0.9907	0.0538	1.949	0.9980	1.854	150
GB-A	RB5	50	0.0581	4.550	0.9920	0.0182	4.949	0.9971	4.653	150
		200	0.0504	8.658	0.9921	0.0078	9.526	0.9976	8.823	120
	RY84	50	0.0446	4.707	0.9840	0.0119	5.239	0.9990	4.878	180
		200	0.0644	8.086	0.9764	0.0114	8.773	0.9986	8.408	150
GB-E	RB5	50	0.0350	3.806	0.9965	0.0102	4.345	0.9961	3.854	150
		200	0.0653	6.692	0.9933	0.0143	7.237	0.9976	6.899	120
	RY84	50	0.0491	2.853	0.9698	0.0226	3.149	0.9969	2.985	180
		200	0.0663	5.285	0.9544	0.0183	5.726	0.9923	5.580	150
GB-EA	RB5	50	0.1878	8.209	0.9839	0.0528	8.481	0.9967	8.533	120
		200	0.1856	32.27	0.9884	0.0140	33.27	0.9984	33.17	90
	RY84	50	0.2079	8.522	0.9934	0.0688	8.731	0.9995	8.692	120
		200	0.2801	30.41	0.9989	0.0441	30.75	0.9999	30.69	60

**Table 4 materials-16-04586-t004:** Dye diffusion rate constants, determined using the intraparticle diffusion model.

Sorbent	Dye	Dye Conc.	Phase 1			Phase 2		
			k_d1_	Duration	R^2^	k_d2_	Duration	R^2^
		[mg/L]	[mg/(g*min^0.5^)]	[min]	-	[mg/(g*min^0.5^)]	[min]	-
GB	RB5	50	0.1379	45	0.9898	0.0241	105	0.9797
		200	0.2325	30	0.9981	0.0972	90	0.9767
	RY84	25	0.1791	30	0.9907	0.0253	150	0.9545
		200	0.3012	20	0.9999	0.0464	130	0.9246
GB-A	RB5	50	0.6231	45	0.9881	0.0861	105	0.9599
		200	1.2292	30	0.9997	0.3751	90	0.9696
	RY84	50	0.6322	30	0.9997	0.1452	150	0.9603
		200	1.3567	20	0.9992	0.2898	130	0.9651
GB-E	RB5	50	0.4546	45	0.9921	0.1516	105	0.9625
		200	1.0405	30	0.9966	0.1829	90	0.9180
	RY84	50	0.3977	30	0.9936	0.1140	150	0.9832
		200	0.8971	20	0.9862	0.2103	130	0.9799
GB-EA	RB5	50	≥2.2602	≤10	0.(9)	0.1512	110	0.9596
		200	≥8.8253	≤10	0.(9)	0.7212	80	0.9499
	RY84	50	≥2.3941	≤10	0.(9)	0.1302	110	0.8919
		200	≥9.0628	≤10	0.(9)	0.3889	50	0.9082

**Table 5 materials-16-04586-t005:** Constants determined from Langmuir 1, Langmuir 2, and Freundlich models.

Sorbent	Dye	Langmuir 1 Model			Langmuir 2 Model						Freundlich Model		
		Q_max_	K_c_	R^2^	Q_max_	b_1_	K_1_	b_2_	K_2_	R^2^	k	n	R^2^
		mg/g	L/mg	-	mg/g	mg/g	L/mg	mg/g	L/mg	-	-	-	-
GB	RB5	2.32	0.0188	0.9983	2.32	0.91	0.0188	1.41	0.0188	0.9983	0.198	2.347	0.9523
	RY84	2.27	0.0229	0.9960	2.27	1.14	0.0229	1.13	0.0229	0.9960	0.240	2.538	0.9357
GB-A	RB5	10.62	0.0313	0.9972	10.62	5.31	0.0313	5.31	0.0313	0.9972	1.245	2.545	0.9402
	RY84	9.85	0.0344	0.9965	9.85	4.90	0.0344	4.95	0.0344	0.9965	1.300	2.688	0.9237
GB-E	RB5	8.16	0.0291	0.9945	8.17	0.28	0.0179	7.89	0.0295	0.9985	0.954	2.577	0.9469
	RY84	6.81	0.0239	0.9978	6.93	0.42	0.1209	6.51	0.0209	0.9980	0.687	2.451	0.9586
GB-EA	RB5	71.30	0.0242	0.9979	71.30	14.30	0.0242	57.00	0.0242	0.9979	6.115	2.257	0.9560
	RY84	59.28	0.0281	0.9992	59.28	29.73	0.0281	29.55	0.0281	0.9992	6.364	2.506	0.9476

**Table 6 materials-16-04586-t006:** Comparison of the sorption properties of various biosorbents and activated carbons towards RB5 and RY84 dyes.

Dye	Sorbent	Sorption Capacity [mg/g]	pH of Sorption	Time of Sorption [min]	Source
RB5	Activated carbon (powder)	125.79	2	240	[14]
	Aminated wheat straw (activated with epichlorohydrin)	91.04	3	210	[20]
	Aminated buckwheat hulls (activated with epichlorohydrin)	85.18	3	300	[23]
	Aminated goldenrod biomass (activated with epichlorohydrin)	71.30	3	120	This work
	Modified activated carbon SPC	69.90	2	<60	[54]
	Powdered activated carbon	58.82	-	-	[55]
	Aminated sunflower seed shells (activated with epichlorohydrin)	51.00	3	240	[22]
	Activated carbon from bamboo	39.02	2	60	[56]
	Activated carbon from carob tree	36.90	2	120	[44]
	Aminated cotton fibers (activated with epichlorohydrin)	36.77	3	240	[21]
	Rape stalks (waste)	32.80	2.5	30	[57]
	Banana peel (powder)	26.90	3	60	[58]
	Activated carbon from palm shell	25.10	2	300	[59]
	Wood (walnut) activated carbon	19.30	5	400	[60]
	Aminated wheat straw (without activation)	17.45	3	210	[20]
	Wheat straw	16.72	3	210	[20]
	Wheat straw (other research)	15.70	7	195	[61]
	Beech sawdust	13.90	3	1440	[40]
	The seed scales of Eriobotrya japonica	13.76	3	150	[43]
	Cotton seed husks	12.90	2	30	[62]
	Aminated goldenrod biomass (without activation)	10.62	3	150	This work
	Aminated buckwheat hulls (without activation)	7.41	3	300	[23]
	Buckwheat hulls	4.43	3	300	[23]
	Sunflower seed	2.89	3	210	[22]
	Cotton fibers	2.74	3	240	[21]
	Goldenrod biomass	2.32	3	150	This work
	Macadamia seed husks	1.21	3	510	[63]
	Sunflower biomass	1.10	2	210	[64]
	Pumpkin seed husks	1.00	3	60	[47]
	Coconut shells	0.82	2	60	[37]
RY84	Aminated sunflower seed hulls (activated with epichlorohydrin)	63.30	3	240	[22]
	Aminated goldenrod biomass (activated with epichlorohydrin)	59.28	3	120	This work
	Aminated cotton fibers (activated with epichlorohydrin)	43.34	2	240	[21]
	Activated carbon from the Borassus flabellifer plant	40.00	-	-	[65]
	Cotton fibers	15.90	2	240	[21]
	Wool	11.00	7	180	[45]
	Aminated goldenrod biomass (without activation)	9.85	3	180	This work
	Sunflower seed husks	4.15	2	90	[22]
	Goldenrod biomass	2.27	3	180	This work
	Compost	2.20	3	180	[38]

## Data Availability

Not applicable.

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
