# Peer review of "The Effect of Modifying Canadian Goldenrod (Solidago canadensis) Biomass with Ammonia and Epichlorohydrin on the Sorption Efficiency of Anionic Dyes from Water Solutions"

_materials, 2023, doi:10.3390/ma16134586_

Round 1

Reviewer 1 Report

1. Lines 9 & 118. The scientific name of a species should be spelt in italic.

2. Abstract. Provide a short sentence on the conclusion of this work. 

3. Line 124. State the purity of both dyes and their solubility in water (in g/ml).

4. Sections 2.3 and 2.4 would be better to be written in paragraphs.

5. Lines 138 & 173. State the unit for 30% ammonia in v/v

6. Line 115. The correct word should be UV-Visible

7. Line 171. What is d.m?

8. Line 212. Rephrase this sentence.

9. Line 233. Spelling error 'The The.

10. Equations 2-7. Please write the equations properly using the Microsoft Word equation functions. 

11. Line 245. The correct term should be intraparticle diffusion.

12. Equation 7 should be Q = K.C1/n. Recalculate the values on n in Table 5

13. Line 283-284. The asymmetric stretching should be at 2919 cm-1. Symmetric stretching at 2849 cm-1. 

14. Figure 2. Spectra GB-A and GB-EA seem to have a reduction in peak intensity at 1730 cm-1. Please explain. 

15. Line 539. Chemisorption often involves electron donor and electron acceptor groups. How could this interaction be possible with these sorbents and adsorbates? Is there any possibility that electrostatic interaction, pi-pi interaction and n-pi interaction are involved during sorption?

16. Sections 3.3 and 3.4 lack references to support the authors' findings/explanation. Please add relevant citations.

17. Any reasons for not conducting the desorption and regeneration studies of the sorbents?

18. Figure 6. The legends should be properly placed so that they do not interfere with the plots.

19. I can't find the procedure for pHpzc determination in the methodology section. 

Some spelling mistakes are noticed in the manuscript. A few sentences need to be rephrased or rewritten for clarity.  

Author Response

Reviewer 1

Reviewer comments

Authors response

1. Lines 9 & 118. The scientific name of a species should be spelt in italic.

Authors are grateful for this remark. Appropriate correction has been made in revised he manuscript.

2. Abstract. Provide a short sentence on the conclusion of this work.

Following Reviewer’s suggestion, a conclusion has been added to Abstract.

Amination of biomass pre-activated with epichlorohydrin can increase its sorption capacity even several dozen times.

3. Line 124. State the purity of both dyes and their solubility in water (in g/ml).

Following Reviewer’s suggestion, information about dye purity and solubility in water has been completed in Table 1.

4. Sections 2.3 and 2.4 would be better to be written in paragraphs.

Authors appreciate this remark.

Of course, it is possible to present chemical reagents and laboratory equipment used in the study in paragraphs. However, we do believe that their presentation in points is more comprehensible and convenient to potential readers.

5. Lines 138 & 173. State the unit for 30% ammonia in v/v

Producer of this chemical reagent did not provide the volumetric unit, but provided information about solution density – 0.892 g/mL. This information has been completed in the revised text.

Based on density of the ammonia + water solution at a temperature 25oC, it is however possible to compute that the volumetric ratio of ammonia to the total solution volume is 37.37 %.

6. Line 115. The correct word should be UV-Visible

Authors are grateful for this remark.

Presumably, the Reviewer meant line 155 (because text in line 115 did not refer in any way to UV-VIS).

Text has been completed with the following information.

“UV-3100 PC - UV/Visible spectrophotometer”

7. Line 171. What is d.m?

It is an abbreviation of “dry matter”.

However, “DM” seems more appropriate as a “dry matter” acronym.

Text has been respectively corrected.

8. Line 212. Rephrase this sentence.

After: 0, 10, 20, 30, 45, 60, 90, 120, 150, 180, 210, 240, and 300 min, 5-mL samples of the solutions were collected with an automatic pipette to the earlier prepared test tubes.

Following Reviewer’s suggestion, the sentence has been rephrased as follows.

Samples of the solutions (5 mL each) were collected by an automatic pipette at intervals of 0, 10, 20, 30, 45, 60, 90, 120, 150, 180, 210, 240, 270, and 300 min into the previously prepared test tubes.

9. Line 233. Spelling error 'The The.

Text has been corrected as suggested by the Reviewer.

10. Equations 2-7. Please write the equations properly using the Microsoft Word equation functions.

Equations have been corrected following Reviewer’s suggestion.

11. Line 245. The correct term should be intraparticle diffusion.

Text has been corrected as suggested by the Reviewer.

12. Equation 7 should be Q = K.C1/n. Recalculate the values on n in Table 5

Equation and calculation results have been corrected in Table 5, as suggested by the Reviewer.

13. Line 283-284. The asymmetric stretching should be at 2919 cm-1. Symmetric stretching at 2849 cm-1.

Text has been corrected as suggested by the Reviewer.

14. Figure 2. Spectra GB-A and GB-EA seem to have a reduction in peak intensity at 1730 cm-1. Please explain.

The lack of peaks at 1730 cm-1 in GB-A and GB-EA spectra has been explained in the revised text.

“The lack of these peaks in GB-A and GB-EA spectra may suggest that the carboxyl and carbonyl groups entered into reactions with ammonia during chemical modification of the test material.”    

15. Line 539. Chemisorption often involves electron donor and electron acceptor groups. How could this interaction be possible with these sorbents and adsorbates? Is there any possibility that electrostatic interaction, pi-pi interaction and n-pi interaction are involved during sorption?

Chemisorption consists in binding sorbent molecules via a chemical reaction and formation of strong covalent bonds between the sorbent and the sorbate.

In the case of GB-E, it is feasible when the sorbent possesses reactive epoxide groups capable of entering into reactions with, e.g., amine or hydroxyl groups of dyes and attaching them to sorbent’s structure by way of addition.

Of course, as mentioned in the manuscript, the major mechanism of dye sorption involves electrostatic interactions (especially at low pH). The pi-pi or n-pi interactions are also likely.

16. Sections 3.3 and 3.4 lack references to support the authors' findings/explanation. Please add relevant citations.

Following Reviewer’s suggestion, respective literature references have been added in sections 3.3-3.4 to support authors’ theories.

17. Any reasons for not conducting the desorption and regeneration studies of the sorbents?

Investigations addressing regeneration of Canadian goldenrod biomass are economically non-viable.

1.      Goldenrod biomass is a very cheap sorptive material. Its regeneration is, therefore, economically ineffective.

2.      Goldenrod biomass structure is relatively unstable and, presumably, the sorbent would be damaged in the successive sorption and regeneration cycles.

3.      Regeneration involving dye desorption generates wastewater that needs to be treated/disposed. In addition, it requires chemical reagents, which generates additional costs.

In authors’ opinion, a viable means for managing spent sorbents from biomass is their drying and co-combustion in, e.g., heating plants at temperatures > 850oC (energy recovery). Production of activated carbons via carbonization and activation of goldenrod biomass is also an option.

18. Figure 6. The legends should be properly placed so that they do not interfere with the plots.

Legends in Figure 6 have been corrected, as suggested by the Reviewer.

19. I can't find the procedure for pHpzc determination in the methodology section. 

Following Reviewer’s suggestion, the procedure of pHPZC determination has been completed in the revised manuscript.

2.9.1. Measurement of pHPZC using the “drift” method

            In research on the pHPZC of sorbents, deionized water with pH correction to the pH range of 2-11 was used instead of dye solutions. After 2h of mixing the sorbent in water solutions (pH 2-11), its pH was measured. A line chart was made (the X-axis is the initial pH, and the Y-axis is the difference between the final pH and the initial pH (pHE-pH0)) for each sorbent. The intersection of the line with the X axis denotes the pHPZC point of the sorbent.

Reviewer 2 Report

This paper reports on the possibility of enriching Canadian goldenrod biomass with amine functional groups to enhance its effect on the sorption efficiency of popular industrial dyes, namely, Reactive Black 5 and Reactive Yellow 84. Results show that amination of Canadian goldenrod biomass can significantly increase its sorption capacity towards anionic dyes.

The proposed method was validated through analyses of the effect of pH on the dye sorption efficiency, of the dye sorption kinetics and of the maximal sorption capacity of the sorbents.

This topic and the proposed application is of relevance nowadays and new applications in removing dyes from waste waters can be envisaged.

Overall, the structure of the work and the manuscript is correct. The introduction section is well structured with adequate number and use of references. In the Results and Discussion section, the authors make adequate use of references to works done by other authors to interpret the obtained results. In this section, the authors correctly describe the experimental work for the proposed new method.

The only criticism that I can see with this manuscript is the fact that no mechanical testing was performed to evaluate the influence of the amination on the mechanical properties of the Canadian goldenrod. But I suspect that those tests are out of the scope of this work.

It is my opinion that this work should be published in Materials as is, since I did not find any major issue.

Author Response

Reviewer 2

This paper reports on the possibility of enriching Canadian goldenrod biomass with amine functional groups to enhance its effect on the sorption efficiency of popular industrial dyes, namely, Reactive Black 5 and Reactive Yellow 84. Results show that amination of Canadian goldenrod biomass can significantly increase its sorption capacity towards anionic dyes.

The proposed method was validated through analyses of the effect of pH on the dye sorption efficiency, of the dye sorption kinetics and of the maximal sorption capacity of the sorbents.

This topic and the proposed application is of relevance nowadays and new applications in removing dyes from waste waters can be envisaged.

Overall, the structure of the work and the manuscript is correct. The introduction section is well structured with adequate number and use of references. In the Results and Discussion section, the authors make adequate use of references to works done by other authors to interpret the obtained results. In this section, the authors correctly describe the experimental work for the proposed new method.

Reviewer comments

Authors response

The only criticism that I can see with this manuscript is the fact that no mechanical testing was performed to evaluate the influence of the amination on the mechanical properties of the Canadian goldenrod. But I suspect that those tests are out of the scope of this work.

It is my opinion that this work should be published in Materials as is, since I did not find any major issue.

Authors are very grateful for this positive review.

We plan to investigate the impact of biomass amination on its stability and mechanical properties in future studies.

Reviewer 3 Report

Comments:

The work investigated the possibility of enriching Canadian goldenrod biomass with amine functional groups and its effect on the sorption efficiency of popular industrial dyes: Reactive Black 5 and Reactive Yellow 84. Based on the novelty and quality of this work, I think the manuscript can be accepted for publication after major revision and performed some additional experiments. The main comments are as follows:

1. The authors should investigate the spent sorbent after the adsorption process.

2. The authors need to check the reusability of the samples.

3. The authors need to check the specific surface area of all samples.

4. How to confirm the formation of the interaction between the functional group of sorbent and anionic dyes?

No

Author Response

Reviewer 3

The work investigated the possibility of enriching Canadian goldenrod biomass with amine functional groups and its effect on the sorption efficiency of popular industrial dyes: Reactive Black 5 and Reactive Yellow 84. Based on the novelty and quality of this work, I think the manuscript can be accepted for publication after major revision and performed some additional experiments. The main comments are as follows:

Reviewer comments

Authors response

1. The authors should investigate the spent sorbent after the adsorption process.

Authors appreciate this remark.

The complete characteristics of the spent sorbent would certainly be necessary in studies addressing its management or re-use.

The present study aimed to demonstrate the feasibility of increasing sorption capacity of Canadian goldenrod biomass by its amination (and this goal has been achieved in authors’ opinion).

It was not our intention to study the spent sorbent and its management possibilities.

Certainly, it is an area ripe for further study. Currently, we are carrying out a study into the possibility of managing spent sorbents based on biomass (including, e.g., energy recovery by combustion or biogas production during digestion).

Findings from this study will be published in another manuscript.

2. The authors need to check the reusability of the samples.

Authors are grateful for this comment.

It is generally assumed that plant biomass is a cheap material that proves well as ‘one-time’ sorbent.

Canadian goldenrod biomass is relatively unstable material and would be damaged in the successive regeneration and sorption cycles.

In addition, its regeneration would be cost-ineffective due to its low price.

Furthermore, regeneration involving dye desorption generates a new batch of wastewater that needs to be treated and requires using chemical reagents, which generates additional costs.

As mentioned in the response to Reviewer 1’s remarks, in authors’ opinion, a viable means for managing spent sorbents from biomass is their drying and co-combustion in, e.g., heating plants at temperatures > 850oC (energy recovery). Production of activated carbons via carbonization and activation of goldenrod biomass is also an option.

3. The authors need to check the specific surface area of all samples.

Authors appreciate this comment.

Specific surface area is one of the important parameters of sorbents based on carbons and mineral sorbents. In many cases, a correlation may be observed between the specific surface area and sorbent’s sorptive capacity.

A typical trait of plant biomass is a change of its volume in an aqueous solution. Particle diameters of the 2-3 mm biomass fraction dried at 105oC may increase after a few-minute retention in an aqueous solution. This effect depends, to a great extent, on solution’s pH. The specific surface area of swelling sorbent fragments changes as well. This process additionally modifies the permeability of sorbent cell walls and enhances dye adsorption and absorption capability.

Furthermore, the chemical modification of sorbent with ammonia or epichlorohydrin should not affect changes in sorbent’s surface area.

Hence, there is a very high risk of the lack of any correlation between the specific surface area of the sorbent and its sorption capabilities.

Summing up, the results of measurements of the specific surface area of the tested sorbents (especially of dry biomass prior to its addition to the solution) would be of very limited use. They would not allow to draw any explicit conclusions.

In authors’ opinion, FTIR and pHPZC determinations as well as elementary analysis prove more useful in the present study. These analyses enabled determining the nature of the test material and type of functional groups on its surface (which was of the paramount importance in determining the sorption capability of the analyzed sorbents).

The same scope of studies has been implemented in previous authors’ studies.

https://link.springer.com/article/10.1007/s13762-019-02536-8

https://link.springer.com/article/10.1007/s10570-020-03054-4

4. How to confirm the formation of the interaction between the functional group of sorbent and anionic dyes?

The chemical structure of dyes used in the study is commonly known. In the case of anionic RB5 and RY84 – these are the easily-ionizable sulfone groups.

In the case of the sorbents tested in our study, these are, i.a., hydroxyl groups and amine groups capable of easy protonation (confirmed by FTIR analysis).

The electrostatic interaction between functional groups of sorbents and dyes has been evidenced in figures presenting a correlation between pH and dye sorption efficiency.

The high efficiency of dye sorption at low pH is explained by strong electrostatic attraction between the protonated functional groups on sorbent’s surface and the negatively charged functional groups of dyes. This mechanism is well known and has been extensively described in literature.

Round 2

Reviewer 1 Report

1. Line 134, the value 2 in NH3.H2O should be subscript

Author Response

The authors are grateful for this suggestion.
The text has been corrected.

Reviewer 3 Report

The authors clarified the concerning points. So, It can be accepted as it.

-

Author Response

The authors are grateful to the reviewer for the opinion that the manuscript can be accepted.